# Weekday and Weekend Physical Activity of Preschool Children in Relation to Selected Socioeconomic Indicators

**DOI:** 10.3390/ijerph19094999

**Published:** 2022-04-20

**Authors:** Jarosław Herbert, Piotr Matłosz, Alejandro Martínez-Rodríguez, Krzysztof Przednowek, Muhammad Asif, Justyna Wyszyńska

**Affiliations:** 1Institute of Physical Culture Sciences, Medical College, University of Rzeszów, 35-959 Rzeszów, Poland; pmatlosz@ur.edu.pl (P.M.); krzprz@ur.edu.pl (K.P.); 2Department of Analytical Chemistry, Nutrition and Food Science, Faculty of Sciences, University of Alicante, 03690 Alicante, Spain; amartinezrodriguez@ua.es; 3Govt. Associate College Qadir Pur Raan, Multan 60000, Pakistan; asifmalik722@gmail.com; 4Institute of Health Sciences, Medical College, University of Rzeszów, 35-959 Rzeszów, Poland; jwyszynska@ur.edu.pl

**Keywords:** accelerometry, objective monitoring, young children, physical activity, preschool

## Abstract

Physical activity (PA) is as vital for improving the health of young children as it is positively associated with a broad range of psychological, cognitive, and cardio-metabolic outcomes. The aims of this study were to: (1) to assess the level of PA and meeting the WHO recommendations: moderate-to-vigorous physical activity (MVPA) and the number of steps in Polish preschool boys and girls on weekdays and on weekends; (2) to investigate the relationship between selected socioeconomic indicators (self-reported by parents) and PA, including meeting the WHO recommendation for daily MVPA and the number of steps on weekdays and on weekends among Polish preschoolers. Data were collected in the 2017/2018 school year. The study included a total of 522 boys and girls both aged between 5 and 6 years. The ActiGraph GT3X-BT tri-axial accelerometer was used to measure PA. Selected socioeconomic indicators as well as parental body weight and body height were self-reported by parents/caregivers using a questionnaire. In most of the PA indicators analyzed for girls (moderate, vigorous, total MVPA, and steps/day), the averages were higher during the week than during the weekend. Moreover, significantly more boys met the criteria of MVPA, both on weekdays and over the weekend (32.3% boys and 19.2% girls on weekdays and 31.1% boys and 18.1% girls on weekends). Additionally, more boys met the step recommendations, but only on weekends (15.5% boys and 6.6% girls). It was found that if there were two people in a household, there was an almost a three-fold greater chance (adj. OR = 2.94, *p* = 0.032) of meeting the MVPA criterion with an even stronger association (over fivefold greater chance) in meeting the step recommendation (adj. OR = 5.56, *p* = 0.033). The differences in the day schedule may potentially contribute with the level of PA in girls. Among the analyzed selected socioeconomic indicators, only the number of people in a household had a significant association on PA.

## 1. Introduction

Physical activity (PA) during the preschool years is crucial for child development and health as well as well-being [1,2]. Regular PA during the preschool age protects against the accumulation of excess body fat [3], whereas high amounts of sedentary behavior (SB) are associated with an increased risk of being overweight or obese [4]. Furthermore, regular PA helps the motor, musculoskeletal, social, and psychological development of preschool children [5]. Therefore, it is important to increase PA in early childhood [6,7], and increases in PA intensity allow for new motor experiences during the preschool years [8].

Moreover, PA may have positive correlations regarding brain function, cognitive development, and school performance among young children [9,10,11]. Furthermore, the consolidation of strong PA habits during childhood has positive long-term effects on lifestyle throughout adulthood [12], and the results of a recent systematic review and meta-analysis of longitudinal studies showed that PA is beginning to decline in childhood compared to previous scientific reports [13].

Currently, public health guidelines on PA place considerable emphasis on the population of children aged 6 to 11 years [14]. These guidelines typically address the frequency, timing, and intensity of PA. The World Health Organization (WHO) and public health authorities around the world recommend that children aged 5–17 years should perform moderate-to-vigorous PA (MVPA) for at least 60 min daily for optimal health benefits [14]. In contrast, the Canadian PA guideline for preschool children suggests children aged 3–6 years should participate in at least 180 min of PA to achieve health benefits [15]. However, it is important to remember that all levels of PA, LPA (light-intensity PA), MPA (moderate-intensity PA), and VPA (vigorous-intensity PA) are important [16]. Studies have shown that preschool children spend most of their day on SB and a small proportion of their day on moderate- or vigorous-intensity PA (MVPA) (<15%) [17,18]. The most commonly used measure for PA analyses is MVPA. Moderate-to-vigorous PA has been shown to be essential for health promotion and disease prevention [16,19] and a large proportion of children do not meet the guidelines for minimum time spent on MVPA [20,21,22,23], and the situation worsens with age [24]. Martínez-Bello and Estevan [25], based on an analysis of the latest literature, confirmed factors that impact PA and motor competence are not only present in primary education but are already manifested in early childhood.

Preschool children typically spend a large amount of time with their parents, who can have a strong influence on their behavior, including PA. Research has shown that PA in early childhood is associated with parental practices that encourage or discourage engagement in PA [26,27,28,29]. Additionally, there are inconsistent findings on the association between parents, educational level, and children’s PA [30,31]. Physical activity is influenced by psychological, social, environmental, and demographic variables [32], and analyses of non-modifiable socio-demographic variables such as age and sex influence lower levels of PA in children [33,34].

In addition, Bassul et al. [35] reported that most parents have positive perceptions and high satisfaction with all aspects of their neighboring environment, such as it being perceived as pleasant and safe for walking and cycling, the number of PA facilities, and the quality and availability of local restaurants and food shops.

Selected socioeconomic status (SES, which commonly refers to educational level, social class, and/or income) have direct or indirect links to various health indicators [36].

Low education and/or the professional skills of parents are one of the most important risk factors for children internalizing (emotional) and externalizing (behavioral) problems [37]. Low SES is associated with low PA [38], and it is also a factor that strongly influences PA [39]. Studies report that youth who are considered to be lower SES participate in less PA than their more advantaged counterparts [40,41]. Additionally, children in lower SES households in the US and other developed countries are more likely to be overweight or obese [42]. Children with lower SES may experience greater barriers to being or becoming physically active. For example, factors of lower household income would affect participation in PA, as those less financially well off cannot afford to participate, resulting in lower average daily PA than children in the high SES group.

For this reason, it is important to consider the various causes and factors that may affect children’s PA levels. A study by Hesketh et al., 2006, showed an association between PA and household income in preschool children [43]. However, systematic reviews showed no relationships between selected socioeconomic indicators and the level of PA in preschool children [44,45]. Among other factors, focusing on an analysis of parent–child relationships in preschool children is the key to understanding the factors that are important in shaping a child’s active lifestyle [12,46]. Nonetheless, PA studies using objective methods are in the minority, and there is currently a lack of valuable studies of objective measurements of parent–child PA in the context of material conditions and other selected factors (e.g., the number of people in the family).

A variety of mechanisms, including encouragement, beliefs, and attitudes towards PA, role modeling, and involvement can help to shape important attitudes and behaviors associated with PA in children [47]. Family members and parents have a strong influence on PA [48], which can take many forms, such as modeling and encouragement [49], rules and restrictions [50], participation [51], and in watching or supervising [52].

In addition, researchers report conflicting findings regarding children’s PA on weekends and weekdays. The day of the week is an important determinant of MVPA, as young children are more sedentary and less physically active [44,53]. Brooke et al. [54], Lee et al. [55], and Gråstén et al. [56] reported in their works that children are more engaged in MVPA on weekdays than during weekends. In contrast, Nupponen et al. [57], Hinkley et al. [45], and Van Cauwenberghe et al. [58] indicated that children were more often physically active on weekends rather than on weekdays. Children’s PA levels on weekends are important because children may have more time to engage in outdoor play and recreational activities compared to weekdays when they spend many of their hours in a seated position. Therefore, it is necessary to study children’s PA during both weekends and weekdays to better understand behaviors and to promote children’s PA.

Taking into account the fact that there is a lack of current and wide-ranging studies among Polish preschoolers that compare socioeconomic indicators with objectively measured PA, the present study focuses on the assessment of these relationships.

To the best of our knowledge, this is the first study to examine the associations between a device-based measure of PA, the number of steps and meeting recommendations for MVPA, and selected socioeconomic indicators on weekdays and on weekends. Therefore, the outcomes of present study may be used in the development of preventive programs addressed to the pediatric population.

The aims of this study were: (1) to assess the level of PA and meeting the WHO recommendations (MVPA and the number of steps) in Polish preschool boys and girls on weekdays and on weekends; (2) to investigate the relationship between the selected socioeconomic indicators (self-reported by parents) and PA, including meeting the WHO recommendation for daily MVPA and the number of steps on weekdays and on the weekends among Polish preschoolers. We hypothesize that there is a difference in PA level and meeting the WHO recommendation for daily MVPA and the number of steps on weekdays and weekends among Polish children in kindergartens. We also assume that in this group there is an association between selected socioeconomic indicators and the abovementioned variables related to PA.

## 2. Materials and Methods

This study was approved by the Bioethics Committee of the University of Rzeszów (no. 2017/01/05) and was conducted in accordance with the ethical standards stated in the relevant version of the Declaration of Helsinki. Before the study was initiated, written consent for participation was obtained from the children’s parents.

### 2.1. Procedures

Based on data published by the Polish Central Statistical Office, there were approximately 6800 preschoolers between 2017 and 2018 in Rzeszów, Poland. Assuming a confidence level of 95% and a 5% margin of error, the required sample size should include at least 364 participants. The invitation to participate in the study was sent by researchers to all kindergartens in Rzeszów. The consent of 41 kindergarten principals were obtained for participation in this study. Based on data from Rzeszów City Hall, the average number of preschoolers attending each kindergarten is approximately 70. Considering an estimate that approximately 30% of parents would agree for child participation in the study, and possible complications during examinations (missing data in surveys, failure to meet valid days when measuring PA with accelerometers, absence of children on the day of the test, etc.), we decided to randomly select 22 kindergartens for participation (kindergartens were selected using STATISTICA software—sampling without replacement). The consent form, the questionnaires, and detailed guidelines were delivered to parents of all children attending to selected kindergartens through kindergarten staff. The parents were instructed to discuss and complete the questionnaire together. In case of two or more children from one family, each child received their own code, and the parents were instructed to complete separate questionnaires for each one.

The signed consents and filled questionnaires (*n* = 565) were collected by kindergarten staff. With the help of tutors/teachers, all participants received comprehensive information about the study.

Anthropometric measurements were carried out in kindergartens, in a separate room. All the measurements were taken between 8:00 a.m. and 10:00 a.m.

### 2.2. Participants

The inclusion criteria consisted of children (1) who were 5 to 6 years old, (2) whose parents or caregivers provided written parental consent and child assent prior to data collection, and (3) attending preschool in Rzeszów. Exclusions criteria were: (1) any conditions that may affect the assessment of PA.

During the data collection stage, 43 subjects were excluded from the study for the following reasons: strong anxiety of examination (*n* = 5), failure to return or complete the survey (*n* = 20), and a lack of valid accelerometer data (*n* = 18). Therefore, to the final analysis, 522 children were included (Figure 1).

The final sample consisted of 522 preschoolers (271 girls) aged 5–6 years (5.4 ± 0.6).

### 2.3. Anthropometric Measurements

Body height was measured to an accuracy of 0.1 cm using a portable stadiometer (Tanita HR-200, Tokyo, Japan). This measurement was taken in a vertical position, barefoot. Bodyweight was assessed with an accuracy of 0.1 kg using a body composition analyzer (BC-420 MA, Tanita, Tokyo, Japan) [59].

Body mass index (BMI) was calculated as body weight (in kg) divided by height in meters squared (kg/m^2^). Based on the BMI values, BMI percentiles were calculated by referencing Polish centile charts [60]. Based on the BMI percentile values, categories of participant body mass were determined as follows: underweight (<5th percentile), healthy body mass (between 5th and 85th percentile), overweight (BMI ≥ 85th percentile and <95th percentile), or obese (≥95th percentile) [61].

### 2.4. Physical Activity

The ActiGraph GT3X-BT tri-axial accelerometer (ActiGraph, Pensacola, FL, USA) was used to measure PA. Currently, accelerometers are used in many studies on the level of PA [62]. Actigraphy is a valid method to objectively measure PA level in preschool children [63].

The accelerometer was worn at the participant’s right hip. After the end of the recording, the sensor was connected to a computer via a mini-USB for data transfer. The participants were instructed to wear the accelerometer for seven consecutive days, 24 h a day, five days a week, and during the two days of the weekend. The data were collected in 5-s epochs [64]. Non-wear time was defined as 60 min of consecutive zeros, allowing for 2 min of non-zero interruptions [65].

Wear time of ≥500 min/day was used as the criterion for a valid day, and ≥4 days were used as the criteria for a valid 7-day period of accumulated data [58]. ActiGraph data were analyzed with Actilife 6.13. (ActiGraph LLC, Pensacola, FL, USA). The cut-off points from Evenson et al. were selected to determine the time spent on MVPA level (>2296 counts per minute—CPM). The cut-off points were: Sedentary: 0–100 CPM, Light: 101–2295 CPM, Moderate: 2296–4011 CPM, Vigorous: 4012–∞ CPM [66]. The participants complying with the minimum 60 min of MVPA per day requirement met the guidelines, while the participants who did not meet this number (<60 min) were regarded as inactive [67]. Physical activity values were compared with the established recommendations of ≥60 min of MVPA or ≥180 min of PA [68] at any intensity to evaluate the proportion of participants meeting these recommendations. Daily step count was calculated as the mean daily step count from all valid days. All step counts below 1000 and above 30,000 steps per day were deleted and treated as missing data according to the rules of Rowe et al. [69]. Participants with at least 12,000 steps per day were considered to be sufficiently physically active [70].

### 2.5. Selected Socioeconomic Indicators

The parents of the participating children were given a questionnaire that they were asked to complete to provide relevant information, including their education level and family structure. Selected socioeconomic indicators and socio-demographic characteristics (children’s sex and date of birth, place of residence, number of people in household, and parents’ education), as well as parental body weight and body height, were self-reported by the parents/caregivers using a questionnaire. Based on parents’ answers in the questionnaire, household income factors were defined as low, middle, or high. Parental BMI was calculated as underweight (BMI < 18.5), normal (BMI 18.5–24.9), overweight (BMI 25–29.9), and obese (BMI ≥ 30) [71].

### 2.6. Statistical Analysis

Statistical analysis was performed using SPSS 20 software (IBM, North Harbour, UK). The data were presented as the mean ± standard deviation (SD) and percentage (%) for continuous and categorical variables, respectively. Univariate and multivariate logistic regression analysis was performed to assess the significant determinants of meeting the MVPA criterion (>60 min/day) according to BMI percentiles and socioeconomic factors. The covariates included sex, various indicators of PA (np. MVPA), and the number of steps. The McNemar test was used to determine the significance of differences in MVPA and steps per day on weekdays vs. weekends. The Wilcoxon signed rank test was used to compare the two measurement variables. The normality of the distribution was applied, and the non-parametric Wilcoxon test was used. The level of statistical significance was set at *p* < 0.05. For measuring the effect size, the value of eta square (η^2^) was calculated and interpreted using the following criteria: no effect (η^2^ < 0.01), small effect (0.01 ≤ η^2^ < 0.06), moderate effect (0.06 ≤ η^2^ < 0.14), strong effect (η^2^ ≤ 0.14) [72].

## 3. Results

The final sample consisted of 522 preschoolers (271 girls) aged 5–6 years (5.4 ± 0.6). The average height was 116.3 ± 6.0 cm, and the average weight was 21.2 ± 3.6 kg. All participating children were white Caucasian, which is representative of the ethnic demographics of Poland.

Table 1 shows that the overall prevalence of overweight and obesity, defined by BMI, was 6.5% and 3.4%, respectively. Based on BMI in boys, 5.2% were overweight and 3.6% were obese. Similarly, in girls, 7.7% were overweight and 3.3% were obese. There were no significant differences in the occurrence of individual body mass categories between boys and girls.

The place of residence (95% of the respondents live in a city), father’s and mother’s education, were accessed. Differences were found in the mother’s education as a parent of five- and six-year-olds (*p* < 0.037). A higher percentage of mothers declared higher education than fathers. Significant differences were also found between five- and six-year-olds regarding the number of persons in the household (*p* < 0.021). Significantly more six-year-olds (6.9%) were in a two-person family compared to five-year-olds (1.5%). Most parents declared a high household income (92.3%). There were no significant differences between sex.

Table 2 shows the individual levels of PA during weekdays vs. weekends, analyzed separately for groups of girls and boys, and in the total sample. In most of the PA indicators analyzed in girls (moderate, vigorous, total MVPA and steps/day), averages were higher during the week than during the weekend. However, no size effects were observed.

Table 3 shows the distribution of selected parameters relating to PA. The data are presented with regard to age and sex. On both weekdays and weekends, most indicators of PA (light, moderate, MVPA, and steps/day) were higher in boys than girls (*p* < 0.05). Moreover, significantly more boys compared to girls met the criteria of MVPA and steps over the weekend (31.1% vs. 18.1% and 15.5% vs. 6.6%, respectively). Moreover, more boys meet steps recommendations, but only on weekends.

Table 4 compares the results related to meeting the recommendation for PA level on the weekend vs. weekday. Results shows that 45.3% of girls and 55.4% of boys met MVPA recommendations on both weekend and weekdays. The criterion of 12,000 steps at the weekend and on weekday was met by 44.4% boys and girls.

Table 5 presents univariate and multivariate logistic regression analysis assessing the relationship between meeting the MVPA recommendation (≥60 min/day) and selected socioeconomic indicators. It was found that if there were two people in household, there was an almost a three-fold greater chance (adj. OR = 2.94, *p* = 0.032) of meeting the MVPA criterion (≥60 min/day).

A similar situation is found in Table 6, where the univariate and multivariate logistic regression analysis was conducted to assess the significant determinants of meeting the number of steps criterion (≥12,000) depending on selected socioeconomic indicators. It was found that if there were two people in a household, there was more than a fivefold chance (adj. OR = 5.56, *p* = 0.033) of meeting the criteria of steps (≥12,000).

## 4. Discussion

The aim of this study was to analyze children’s PA on both weekends and weekdays to better understand behavior and promote the PA of children, and to analyze the relationship between selected socioeconomic indicators and PA. The results confirmed the hypothesis that Polish preschoolers are potentially more active on weekdays than on weekends. Only 8.4% of participants during the week and 10.9% of participants on the weekend met international guidelines for daily steps. The MVPA score was much better, where the guidelines were met by 25.5% of the participants during the week and 24.3% of the participants over the weekend.

Based on the analysis of the self-reported results, boys were more active than girls. During the week, boys showed significantly higher values of PA parameters (MVPA in boys 50.1 ± 22.4 vs. girls 42.4 ± 20.1). During the weekend, boys also showed significantly higher values of PA parameters (MVPA in boys 48.6 ± 29.6 vs. girls 38.62 ± 21.96). Our findings are therefore consistent with the literature that preschool boys are more active than their female counterparts [73,74].

The results of this analysis show that boys and girls accumulate more MVPA on weekdays compared to weekend days. Similar values were obtained by other researchers [75,76,77]. Despite the fact that only 8.4% of participants on weekdays and 10.9% over the weekend met the recommendation of 12,000 steps per day, the results of our study are not significantly different from those of other authors. Considering the number of steps (weekday 8869.97 ± 2174.43; weekend 8587.43 ± 3017.55), our participants are at an average level compared to other results [78,79]. The results of this analysis show that boys and girls accumulate a higher number of steps on weekdays compared to weekend days. Given the differences in favor of activity during weekdays, many analyses confirm this result [78,80,81]. Our results also show sex differences, with boys being at the forefront of daily number of steps (boys 9109.99 ± 2151.98 vs. girls 8647.67 ± 2175.43). This is also evident in other analyses [82,83]. Possible explanations for significant differences between weekday PA and weekend PA may include participation in more physical activities during weekdays (e.g., PE class), or compulsory class activities in which participation in the curriculum activities are deemed compulsory by the kindergarten. However, the influence of segmented activities during kindergarten hours on step counts and other PA indicators are not conclusive because we were unable to provide relevant evidence [77]. Other authors offer an explanation of the difference in meeting recommendations of PA in weekdays and weekends by differences in sleep duration, family activities or parental support [75,76].

Selected socioeconomic indicators are a complex and multifactorial construct with the most common indicators being education level and income level, among others. One of the most important indicators is the level of education because it reflects not only economic factors, but also general and health knowledge, which may be more important for children’s health behavior than, e.g., income or place of residence. Families with higher incomes can afford to use cars to take the child to kindergarten instead of walking or using public transport. This reflects consistency with a recent study that found that children with higher parental income (as one of the strongest demographic factors) favored participation in organized PA, which could possibly be explained by parental support for such activities [84].

The results in Malmo et al. [31] point towards no significant relationship between preschool children’s activity levels at leisure and their parents’ education level.

Subjects, whose parents reported household income factors, met the recommendations of the MVPA (94.9%) and daily steps (100%).

On the other hand, there were more normal weight children who met the MVPA recommendations (80.8%) compared to their parents declaring no obesity (87.9%) and more normal weight children who met the daily steps recommendations (84.6%) compared to their parents declaring no obesity (88.95%). Adult obesity is most often caused by low PA levels, so it is most likely that children with low PA levels drew patterns from low-activity parents [26,85]. It is also interesting to note that every second child who met the MVPA and daily steps recommendations lived in a family of four (MVPA—45.5%) and daily steps (53.8%).

The results partly support the hypothesis that the selected socioeconomic indicators were associated with higher PA. The findings of this study suggest that the number of people living together in the same house has a great influence on the fulfillment of the PA criteria. If there were two people in the household, the chance of meeting the MVPA criterion was almost three times higher (OR = 2.94). On the other hand, when there were two people in the household, the chance of meeting the steps/day criterion was more than five times higher (adj. OR = 5.56). One possible explanation of this association could be that single parents may put greater attention and spent more time being physically active with a child. It is likely that parents/parent are role models for a child’s extracurricular PA. When children find that their parents are actively involved and value PA, they can adopt the values and behaviors themselves.

Physically active parents can provide more support and encouragement to their children to be active and influence their children’s level of activity in this way. There may be a genetic predisposition to PA. For example, in a review, Beunen and Thomis found that heritability rates for practicing PA ranged from 0.35 to 0.83, and children whose parents were active in PA were 1.2–5.8 times more likely to participate in PA than children whose parents were not active in PA [86]. It is important to remember that parents play an important role in promoting healthy and active lifestyles in their children [87,88]. In the case of other selected socioeconomic indicators, the scale of the differences was insignificant.

Parents that attain higher education can take advantage of parenting practices that encourage PA, as they are more aware of the health benefits of PA. To meet the minimum recommendations of PA, among others, one should: increase activity (walking) over short distances, refrain from installing screens/TV in children’s bedrooms, promote non-competitive PA, improve infrastructure in playgrounds offering more physical games, encourage children to be active, and reduce sedentary time in kindergarten and at home. Nevertheless, some limitations of the study should be mentioned. The data are cross-sectional and do not allow for an analysis of cause-and-effect relationships. Consequently, we do not know the direction of the relationships found.

To the best of our knowledge, this is the first study in Poland to examine the complex associations between a device-based measure of PA, meeting recommendations for MVPA (≥60 min/day) and the number of steps (≥12,000) and selected socioeconomic indicators on weekdays and on weekends. The study covered a relatively large sample of preschoolers. One of the strengths of this study is the provision of objective data on children’s PA.

The cross-sectional nature of the methodology applied in the present study and a statistical analysis which does not include the causality and generalizability models should be recognized as important limitations, and therefore our outcomes should be treated with caution. Present research was based mainly on analyzes of PA and selected socioeconomic indicators (including household income factors). Therefore, future research should focus on the full analysis of SES versus PA and sedentary behaviors of children and their parents, both at the weekdays and the weekend. Moreover, the questionnaire used in the study should be considered as a limitation of the study, as it has not been validated. Therefore, results related to the use of this questionnaire should be treated with caution. Another limitation of this study could be that the 7.6% of participants who meet the inclusion criteria were excluded from the analysis due to a strong anxiety about examination, a failure to return or complete the survey, and a lack of valid accelerometer data. The results may be limited to some extent because all kindergartens were located in one metropolitan area. In addition, the research sample was not an ideal representation of the population of children attending kindergarten. It did not include children from rural areas or other towns. However, it is worth noting that there were no interventions in the children’s PA during the study. Future research should target cause-effect relationships and analyze PA more closely during the day, divided between kindergarten attendance and staying at home.

In summary, this study demonstrated that preschools are a strong predictor of PA levels, which supports the importance of the preschool setting for healthy PA behaviors and development.

While some of the observed correlates were non-modifiable (e.g., selected socioeconomic indicators), these findings may be helpful in identifying families who are at higher-risk of promoting an inactive lifestyle to their young children (e.g., number of people in household). This study suggests that future interventions should be aimed at increasing PA in young children by promoting PA and its benefits to parents.

## 5. Conclusions

Approximately one third of boys and one fifth of girls meet the WHO recommendations according to MVPA and steps. Boys were more physically active than girls both during weekdays and on weekends. The differences in the day schedule may potentially contribute with the level of PA in girls. Among the analyzed selected socioeconomic indicators, only the number of people in household had a significant association on PA. The present findings may help to promote future interventions that focus on increasing PA (including MVPA and daily steps) on both weekday and weekends to improve physical development and maintenance of healthy weight in preschool children. Further research is needed to define modifiable determinants of PA in children attending kindergarten.

## Figures and Tables

**Figure 1 ijerph-19-04999-f001:**
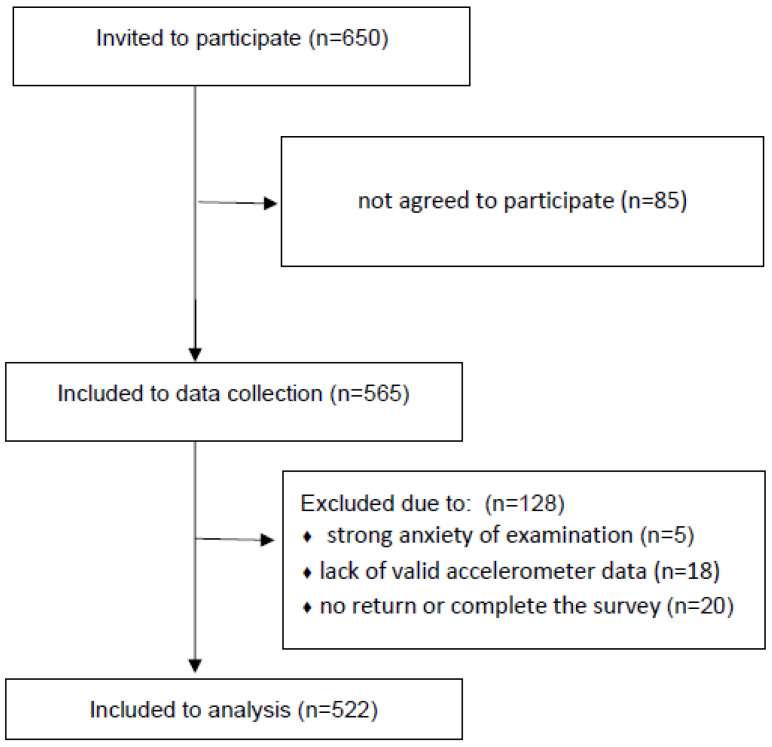
Flow diagram for participant recruitment.

**Table 1 ijerph-19-04999-t001:** Characteristics of the population by age and sex.

Variables	Age (Years)			Sex			Total Sample (*n =* 522)
	5 (*n =* 261)	6 (*n =* 261)	*p* ^a^	Boys (*n =* 251)	Girls (*n =* 271)	*p* ^b^	
Body height (cm) ^†^	116.4 ± 6.1	116.2 ± 5.8	0.761	116.8 ± 6.0	115.8 ± 5.9	0.067	116.3 ± 6.0
Weight (kg) ^†^	21.8 ± 3.6	21.2 ± 3.6	0.859	21.3 ± 3.6	21.0 ± 3.6	0.448	21.2 ± 3.6
BMI (kg/m^2^) ^†^	15.5 ± 1.4	15.6 ± 1.7	0.488	15.5 ± 1.7	15.6 ± 1.6	0.570	15.6 ± 1.6
Body mass classification (according to BMI percentiles) *							
Underweight	22 (8.4)	23 (8.8)	0.809	18 (7.2)	27 (10.0)	0.401	45 (8.)
Normal	215 (82.4)	210 (80.5)		211 (84.1)	214 (79.0)		425 (81.4)
Overweight	17 (6.5)	17 (6.5)		13 (5.2)	21 (7.7)		34 (6.5)
Obesity	7 (2.7)	11 (4.2)		9 (3.6)	9 (3.3)		18 (3.4)
Place of residence *							
Urban	244 (93.5)	252 (96.6)	0.108	243 (96.8)	253 (93.4)	0.074	496 (95.0)
Rural	17 (6.5)	9 (3.4)		8 (3.2)	18 (6.6)		26 (5.0)
Mother’s education *							
Middle school or lower	50 (19.2)	70 (26.8)	**0.037**	60 (23.9)	60 (22.1)	0.632	120 (23.0)
High school/University	211 (80.8)	191 (73.2)		191 (76.1)	211 (77.9)		402 (77.0)
Father’s education *							
Middle school or lower	108 (41.4)	119 (45.6)	0.331	110 (43.8)	117 (43.2)	0.881	227 (43.5)
High school/University	153 (58.6)	142 (54.4)		141 (56.2)	154 (56.8)		295 (56.5)
Number of people in household *							
2	4 (1.5)	18 (6.9)	**0.021**	8 3(3.2)	14 (5.2)	0.662	22 (4.2)
3	64 (24.5)	72 (27.6)		67 (26.7)	69 (25.5)		136 (26.1)
4	142 (54.5)	116 (44.4)		129 (51.4)	129 (47.6)		258 (49.4)
5	35 (13.4)	42 (16.1)		35 (13.9)	42 (15.5)		77 (14.8)
More	16 (6.1)	13 (5.0)		12 (4.8)	17 (6.3)		29 (5.6)
Household income *							
High	241 (92.3)	241 (92.3)	1.00	230 (91.6)	252 (93.0)	0.561	482 (92.3)
Middle or low	20 (7.7)	20 (7.7)		21 (8.4)	19 (7.0)		40 (7.7)
Parental obesity *							
None	224 (85.8)	223 (85.4)	0.126	210 (83.7)	237 (87.5)	0.117	447 (85.6)
Father	11 (4.2)	19 (7.3)		13 (5.2)	17 (6.2)		30 (5.7)
Mother	13 (5.0)	14 (5.4)		19 (7.6)	8 (3.0)		27 (5.2)
Both parents	13 (5.0)	5 (1.9)		9 (3.6)	9 (3.3)		18 (3.4)

Data are expressed as: * *n* (%); ^†^ mean *±* SD; *p*
^a^—the assessment of significance of differences between 5 and 6 years old; *p*
^b^—the assessment of significance of differences between boys and girls; BMI—body mass index; significant associations are highlighted in bold.

**Table 2 ijerph-19-04999-t002:** Levels of physical activity in weekday and weekends.

		Total (*n* = 522)			Girls (*n* = 271)			Boys (*n* = 251)		
Average Physical Activity (min/Day)		Mean ± SD	*p*	η^2^		*p*	η^2^	Mean ± SD	*p*	η^2^
Light	weekday	464.0 *±* 710 456 *±* 97.8 37.9 *±* 17.2 35.5 *±* 20.5 8.3 *±* 6 7.9 *±* 8.7 46.2 *±* 21.6 43.4 *±* 26.3 8873.4 *±* 2183.8 8596.0 *±* 3016.2	0.241	0.002	455.5 *±* 72.5 451.6 *±* 92.6 34 *±* 15.3 31.2 *±* 16.4 8.4 *±* 6.2 7.3 *±* 8 42.4 *±* 20.1 38.5 *±* 21.9 8634.9 *±* 2182.3 8286.9 *±* 2639.4	0.676	0.001	473.3 *±* 68.3 460.8 *±* 103.2 42 *±* 18.1 40.1 *±* 23.3 8.2 *±* 5.7 8.5 *±* 9.4 50.3 *±* 22.5 48.6 *±* 29.5 9130.7 *±* 2160.3 8929.6 *±* 3349.5	0.239	0.005
	weekend									
Moderate	weekday		**0.001**	0.004		**0.002**	0.008		**0.050**	0.002
	weekend									
Vigorous	weekday		**0.001**	0.001		**0.001**	0.005		0.150	0.001
	weekend									
Total MVPA	weekday		**0.001**	0.003		**0.001**	0.008		0.127	0.001
	weekend									
Steps/day	weekday		**0.004**	0.003		**0.011**	0.005		0.148	0.001
	weekend									

η^2^: eta square-effect size; MVPA—moderate to vigorous physical activity; significant associations are highlighted in bold.

**Table 3 ijerph-19-04999-t003:** Physical activity levels on weekdays and during weekends by age and sex.

Variables	Age (Years)			Sex			Total Sample (*n* = 522)
	5 (*n* = 261)	6 (*n* = 261)	*p* ^a^	Boys (*n* = 251)	Girls (*n* = 271)	*p* ^b^	
Weekdays							
Average physical activity on weekday (min/day) ^†^							
Light	463.9 ± 621.1	464.7 ± 78.4	0.908	473.1 ± 68.5	456.1 ± 71.7	**0.006**	464.3 ± 70.6
Moderate	37.0 ± 16.4	38.6 ± 17.8	0.290	41.9 ± 18.0	34.1 ± 15.3	**0.001**	37.8 ± 17.1
Vigorous	8.0 ± 5.7	8.5 ± 6.3	0.391	8.2 ± 5.7	8.3 ± 6.3	0.720	8.3 ± 6.04
MVPA	45.1 ± 20.6	47.1 ± 22.4	0.279	50.1 ± 22.4	42.4 ± 20.1	**0.001**	46.1 ± 21.5
Steps/day	8905.4 ± 2071.9	8834.5 ± 2276.4	0.710	9109.9 ± 2151.9	8647.6 ± 2175.4	**0.001**	8869.9 ± 2174.4
Meeting criteria of MVPA (≥60 min/day) on weekday *							
Yes	63 (24.1)	70 (26.8)	0.482	81 (32.3)	52 (19.2)	**0.001**	133 (25.5)
No	198 (75.9)	191 (73.2)		170 (67.7)	219 (80.8)		389 (74.5)
Meeting criteria of steps (≥12000) on weekday *							
Yes	22 (8.4)	22 (8.4)	0.999	26 (10.4)	18 (6.6)	0.127	44 (8.4)
No	239 (91.6)	239 (91.6)		225 (89.6)	253 (93.4)		478 (91.6)
Weekend							
Average physical activity on weekend (min/day) ^†^							
Light	459.8 ± 871.6	452.6 ± 106.9	0.399	460.9 ± 103.4	451.9± 92	0.292	456.2 ± 97.7
Moderate	34.4 ± 18.2	36.5 ± 22.6	0.256	40.0 ± 23.4	31.2 ± 16.4	**0.001**	35.4 ± 20.5
Vigorous	7.7 ± 8.4	8.1 ± 9.1	0.565	8.5 ± 9.4	7.3± 8	0.103	7.9 ± 8.7
MVPA	42.1 ± 23.7	44.6 ± 28.7	0.282	48.6 ± 29.6	38.6 ± 21.9	**0.001**	43.3 ± 26.4
Steps/day	8731.0 ± 2781.4	8443.8 ± 3235.4	0.277	8907.6 ± 3352.3	8290.8 ± 2642.2	**0.020**	8587.4 ± 3017.5
Meeting criteria of MVPA (≥60 min/day) on weekend *							
Yes	59 (22.60)	68 (26.10)	0.359	78 (31.1)	49 (18.1)	**0.001**	127 (24.3)
No	202 (77.40)	193 (73.90)		173 (68.9)	222 (81.9)		395 (75.7)
Meeting criteria of steps (≥12000) on weekend *							
Yes	29 (11.1)	28 (10.7%)	0.888	39 (15.5)	18 (6.6)	**0.001**	57 (10.9)
No	232 (88.8)	233 (89.)		212 (84.5)	253 (93.4)		465 (89.1)
Total (weekday and weekends)							
Average physical activity in total (min/day) ^†^							
Light	3314.2 ± 553.7	3268.2 ± 457.3	0.302	3279.4 ± 474	3302.2 ± 537.9	0.608 **0.022** **0.026** **0.014** **0.039**	3291.2 ± 507.8
Moderate	256.4 ± 117.3	269.4 ± 113.4	0.198	275.0 ± 116.2	251.8 ± 113.8		262.9 ± 115.4
Vigorous	52.6 ± 36.5	63.0 ± 45.1	**0.004**	62.0 ± 41.7	53.9 ± 40.7		57.8 ± 41.3
MVPA	309.1 ± 143.8	332.4 ± 145.7	0.067	337.0 ± 146.1	305.8 ± 142.8		320.8 ± 145.1
Steps/day	60258.4 ± 14739	64596.8 ± 1482.8	**0.001**	63829.3 ± 14856.1	61129.3 ± 14904.5		62427.6 ± 14928.1
Meeting criteria of MVPA (≥60 min/day) in total *							
Yes	37 (14.2)	62 (23.8)	**0.005**	59 (23.5)	40 (14.8)	**0.011**	99 (19.0)
No	224 (85.8)	199 (76.2)		192 (76.5)	231 (85.2)		423 (81.0)
Meeting criteria of steps (≥12000) in total *							
Yes	5 (1.9)	21 (8.0)	**0.001**	15 (6.0)	11 (4.1)	0.314	26 (5.0)
No	256 (98.1)	240 (92.0)		236 (94.0)	260 (95.9)		496 (95.0)

Data are expressed as: * *n* (%); ^†^ mean *±* SD; *p*
^a^—the assessment of significance of differences between 5 and 6 year olds; *p*
^b^—the assessment of significance of differences between boys and girls; MVPA—moderate to vigorous physical activity; significant associations are highlighted in bold.

**Table 4 ijerph-19-04999-t004:** Meeting physical activity criteria at weekends vs. weekday.

			Total (*n* = 522)			Girls (*n* = 271)			Boys (*n* = 251)		
			(Weekday) Meeting Criteria of MVPA (≥60 min/Day)		Total	(Weekday) Meeting Criteria of MVPA (≥60 min/Day)		Total	(Weekday) Meeting Criteria of MVPA (≥60 min/Day)		Total
			No	Yes		No	Yes		No	Yes	
(weekend) Meeting criteria of MVPA (≥60 min/day)	No	n	330	65	395	194	28	222	136	37	173
		%	85.4	48.5	75.9	88.6	54.7	82.1	81.2	44.6	69.2
	Yes	n	57	70	127	26	22	48	31	48	79
		%	14.6	51.5	24.1	11.4	45.3	17.9	18.8	55.4	30.8
*p*			0.470			0.683			0.630		
			(weekday) Meeting criteria of steps (≥12,000)		Total	(weekday) Meeting criteria of steps (≥12,000)		Total	(weekday) Meeting criteria of steps (≥12,000)		Total
			No	Yes		No	Yes		No	Yes	
(weekend) Meeting criteria of steps (≥12,000)	No	n	441	25	466	244	10	254	197	15	212
		%	92.1	55.6	89.0	96.1	55.6	93.4	87.6	55.6	84.2
	Yes	n	36	20	56	9	8	17	27	12	39
		%	7.9	44.4	11.0	3.9	44.4	6.6	12.4	44.4	15.8
*p*			0.129			1.000			0.066		

MVPA—moderate to vigorous physical activity.

**Table 5 ijerph-19-04999-t005:** Linking MVPA criterion fulfilment parameters (≥60 min/day) to socioeconomic factors.

Variables	Meeting Criteria of MVPA (≥60 min/Day)					
	Yes (*n* = 99) *n* (%)	No (*n* = 423) *n* (%)	Unadjusted OR (95% CI)	*p*	Adjusted OR ^a^ (95% CI)	*p*
Place of residence						
Urban	95 (19.2)	401 (80.8)	REF		REF	
Rural	4 (15.4)	22 (84.6)	0.76 (0.25–2.28)	0.634	0.85 ^a^ (0.28–2.55)	0.774
Mother’s education						
High school/University	72 (17.9)	330 (82.1)	REF		REF	
Middle school or lower	27 (22.5)	93 (77.5)	1.33 (0.81–2.19)	0.261	1.32 ^a^ (0.80–2.17)	0.283
Father’s education						
High school/University	**49 (16.6)**	**246 (83.4)**	**REF**		**REF**	
Middle school or lower	**50 (22.0)**	**177 (78.0)**	**1.42 (0.914–2.201)**	**0.119**	**1.42 ^a^ (0.91–2.21)**	**0.125**
Number of people in household						
2	8 (36.4)	14 (63.6)	2.67 (1.00–7.06)	**0.048**	2.94 (1.10–7.90)	**0.032**
3	24 (17.6)	112 (82.4)	REF		REF	
4	45 (17.4)	213 (82.6)	0.99 (0.57–1.70)	0.950	0.98 (0.57–1.70)	0.946
5	16 (20.8)	61 (79.2)	1.22 (0.60–2.48)	0.574 0.700	1.26 (0.62–2.56) 1.28 (0.47–3.52)	0.528
more	6 (20.7)	23 (79.3)	1.21 (0.45–3.31)			0.629
Household income						
High	94 (19.5)	388 (80.5)	REF		REF	
Middle or low	5 (12.5)	35 (87.5)	0.59 (0.22–1.54)	0.283	0.57 ^a^ (0.22–1.50)	0.265
Parental obesity						
None	87 (19.5)	360 (80.5)	REF		REF	
Father	4 (13.3)	26 (86.7)	0.64 (0.217–1.87)	0.412	0.65 (0.22–1.91)	0.265
Mother	5 (18.5)	22 (81.5)	0.94 (0.34–2.55)	0.904	0.82 (0.30–2.26)	0.715
Both parents	3 (16.7)	15 (83.3)	0.83 (0.23–2.92)	0.769	0.81 (0.23–2.87)	0.755

OR (95% CI)—odds ratio with a 95% confidence interval; REF—reference category; ^a^—the model adjusted for sex; MVPA—moderate to vigorous physical activity; significant associations are highlighted in bold.

**Table 6 ijerph-19-04999-t006:** Linking the parameters of meeting the steps/day criterion (≥12,000) to socioeconomic factors.

Variables	Meeting Criteria of Steps (≥12,000)					
	Yes (*n* = 26) *n* (%)	No (*n* = 496) *n* (%)	Unadjusted OR (95% CI)	*p*	Adjusted OR ^a^ (95% CI)	*p*
Place of residence						
Urban	26 (5.2)	470 (94.8)	REF		REF	
Rural	0 (0.0)	26 (100.0)	0.00 (0.00–0.00)	0.998	0.00 (0.00–0.00)	0.998
Mother’s education						
High school/University	16 (4.0)	388 (96.0)	REF		REF	
Middle school or lower	10 (8.3)	110 (91.7)	2.19 (0.97–4.97)	0.060	2.18 (0.96–4.93)	0.063
Father’s education						
High school/University	12 (4.1)	283 (95.9)	REF		REF	
Middle school or lower	14 (6.2)	213 (93.8)	1.55 (0.70–3.42)	0.285	1.55 (0.70–3.42)	0.280
Number of people in household						
2	3 (13.6)	19 (86.4)	5.21 (1.08–25.10)	**0.040**	5.56 (1.14–26.99)	**0.033**
3	4 (2.9)	132 (97.1)	REF		REF	
4	14 (5.4)	244 (94.6)	1.89 (0.61–5.87)	0.269	1.89 (0.61–5.86)	0.271
5	4 (5.2)	73 (94.8)	1.81 (0.44–7.44)	0.412	1.84 (0.45–7.59)	0.398
more	1 (3.4)	28 (96.6)	1.18 (0.13–10.95)	0.885	1.22 (0.13–11.38)	0.860
Household income						
High	26 (5.4)	456 (94.6)	REF		REF	
Middle or low	0 (0)	40 (100.0)	0.00 (0.00–0.00)	0.998	0.00 (0.00–0.00)	0.998
Parental obesity						
None	23 (5.1)	424 (94.9)	REF		REF	
Father	1 (3.3)	29 (96.7)	0.64 (0.08–4.87)	0.664	0.64 (0.08–4.95)	0.673
Mother	1 (3.7)	26 (96.3)	0.71 (0.09–5.46)	0.741	0.64 (0.08–5.00)	0.675
Both parents	1 (5.6)	17 (94.4)	1.08 (0.14–8.51)	0.941	1.07 (0.14–8.42)	0.948

OR (95% CI)—odds ratio with a 95% confidence interval; REF—reference category; ^a^—the model adjusted for sex; MVPA—moderate to vigorous physical activity; significant associations are highlighted in bold.

## Data Availability

The data is available at the link: https://repozytorium.ur.edu.pl/handle/item/7423 (accessed on 1 July 2021).

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
