# Peer review of "Weekday and Weekend Physical Activity of Preschool Children in Relation to Selected Socioeconomic Indicators"

_ijerph, 2022, doi:10.3390/ijerph19094999_

Round 1

Reviewer 1 Report

This is an interesting study examining the level of intensity of duration of PA in preschool boys and girls on weekdays and at weekends as well as the socio-demographic predictors. I have several comments and suggestions for this manuscript:

  1. The introduction is limited and narrow. The authors should expand the literature on previous studies that examined psychological, socioemotional, and demographic predictors of physical activity.
  2. There is also a need to provide a strong justification why only sociodemographic factors were studied in the current study. Without a strong justification, the current paper seems to be lack of a direction
  3. Hypotheses of the current study should be listed out explicitly at the end of the introduction
  4. Inclusion and exclusion criteria must be explicitly stated in the method section
  5. The number of missing data and its treatment (e.g., listwise, multiple imputation, etc) should be reported.
  6. The justification for the covariates in the adjusted model must be elaborated Statistical Analysis Section
  7. There is a lack of information related to how the authors classified SES. It must be made clearly.
  8. I would appreciate if the material and data is made open access (e.g. on the Open Science Framework) as this will facilitate meta-analysis (which the authors benefit from too
  9. Limitations related to causality and generalizability of the current findings should be elaborated in the Discussion.

Author Response

We are very thankful for your comments about our article and the opportunity to revise our manuscript. We included all your helpful advice and made necessary changes in the article. Herein we explain how we revised the paper based on your comments and recommendations. Changes made in the manuscript are marked using track changes.

Comments for the Authors: The introduction is limited and narrow. The authors should expand the literature on previous studies that examined psychological, socioemotional, and demographic predictors of physical activity.

Author's response: We agree with the reviewer. We modified the introduction section accordingly.

Comments for the Authors: There is also a need to provide a strong justification why only sociodemographic factors were studied in the current study. Without a strong justification, the current paper seems to be lack of a direction

Author's response: Thank you for this suggestion; we agree with the Reviewer; however, taking into account the fact that there is a lack of current, wide-ranging studies among Polish preschoolers that compare socioeconomic indicators with objectively measured PA, we decided to focus on the assessment of these relationships. Moreover, it can be found some studies assessing relationships between other studied on biological risk factors among Polish prescholers. Therefore we decided to focus, in this paper, mainly on socioeconomic indicators.

Comments for the Authors: Hypotheses of the current study should be listed out explicitly at the end of the introduction

Author's response: We agree with the reviewer. We added the hypothesis at the end of the introduction section.

Comments for the Authors: Inclusion and exclusion criteria must be explicitly stated in the method section

Author's response: Thank you for this suggestion. We included the inclusion and exclusion criteria in the Participants subsection.

Comments for the Authors: The number of missing data and its treatment (e.g., listwise, multiple imputation, etc) should be reported

Author's response: We thank the reviewer for this comment. Missing data in present study was mainly due to the following reasons: strong anxiety about examination (n = 5), failure to return or complete the survey (n = 20), or lack of valid accelerometer data (n = 18). Finally, 522 children were included in the analysis (43 subjects were excluded from the study which constitute 8.2% of whole sample). Occurrence of any of the above reasons caused the exclusion from further analysis. We believe that the 7.6% of subjects with missing data should not significantly influence the study outcomes however we consider it as limitation of present study.

Comments for the Authors: The justification for the covariates in the adjusted model must be elaborated Statistical Analysis Section

Author's response: We thank the reviewer for this comment. There were significant differences according to sex in PA, therefore we created an additional model for these variable.

Comments for the Authors: There is a lack of information related to how the authors classified SES. It must be made clearly

Author's response: Thank you for this comment. Due to the incorrect translation in previous version of manuscript, the socio-economical status (SES) term was incorrectly used. Therefore in revised manuscript we had changed the term SES in to „selected socio – economic indicators”. In the present study the selected socioeconomic indicators were: place of residence, education of the parents, number of people in the household and household income.

Comments for the Authors: I would appreciate if the material and data is made open access (e.g. on the Open Science Framework) as this will facilitate meta-analysis (which the authors benefit from too.

Author's response: The link for the data is provided in the data availability statement at the end of manuscript.

Comments for the Authors: Limitations related to causality and generalizability of the current findings should be elaborated in the Discussion.

Author's response: We thank the reviewer for this comment.

The cross-sectional nature of methodology applied in our study and conducted statistical analysis which does not include the causality and generalizability models should be recognised as important limitations, and therefore our outcomes should be treated with caution. 

Reviewer 2 Report

General: Thank you for the opportunity to review this manuscript. This cross-sectional study assessed prevalence of weekday and weekend physical activity in a sample of preschoolers in Poland and its association with familial socio-demographic and socio-economic characteristics. See below my feedback:

Abstract:

The opening statement should be a bit stronger. Why is PA so important at an early age? Even more so “necessary”? Citing some specific benefits might be suitable here.

Pg 1 Ln 17-18: Here you state that you will assess the sample’s PA levels in relation to guideline recommendations. Which guidelines? Also, this should be a step in the context of the first aim (makes most sense).

Since the sample is mostly 5–6-year-olds, I would be more specific when referring to the group. Are these kindergarteners? First graders? In some countries, 5–6-year-olds are no longer considered preschoolers.

Pg 1 Ln 24-25: It is quite interesting that some of your early findings focus on a comparison between 5- and 6-year-olds. Were the authors originally intending to assess such comparisons? At first sight, I would think findings would be summarized for the total sample.

Pg 1 Ln 26: “genders.” Did you mean by biological sex of the child?

Pg 1 Ln 32-38: The discussion provided elaborates on findings not presented. Where are the findings for study aim 2? Authors only presented findings related to guideline recommendations in the abstract. I encourage the authors to re-visit the abstract and be more specific about the indicators and models used to test each study aim.

Introduction:

“Physical activity (PA) during the preschool years is crucial for child development and 42 health as well as well-being [1,2].” Can the authors provide specific examples of developmental and health outcomes that benefit from PA engagement?

Pg 2 Ln 85-87 “Thus, focusing on the analysis of parent-child relationships in preschool children is the key to understanding the factors that are important in shaping children's active lifestyles [3,36].” Was there any assessment of parents’ PA engagement? Even through the questionnaire? When this statement was presented in the introduction, I thought that was where the authors were headed.

Pg 3 Ln 99-105 Same comment as in the abstract: Revise the aims, PA assessment with guidelines should fall under aim 1. Be specific about the vars and models in aim 2.

Methods:

Pg 3 Ln 111-113 “All the measurements were taken between 8:00 a.m. and 10:00 a.m. With the help of tutors/teachers, all participants received comprehensive information about the study.” How was the consent process? Did parents also receive such information? Did the parents provide informed consent? I am also curious about the 2-hour window of the measurement. Is it a validated approach?

Pg 3 Ln 117 “healthy preschoolers.” Please define.

Pg 4 Ln 150-151: How many met criteria for a valid 7-day period of accumulated data? The total sample?

Results

Table 1: The cut-offs for SES and education do not provide insight into how these variables may play a role in your outcome of interest. Perhaps it might be worth considering other theoretically relevant categorizations of SES if relevant to your research question.

Table 2: You have some really interesting findings related to weekday vs weekend PA. That’s the first time in the paper I read this. Should be presented in the abstract (it is not there currently).

The findings about the exact number of people in the household (2) associated with increased odds of meeting guideline recommendations is interesting. First, why 2 people? Is this perhaps a matter of individuals occupying space in the household, while also having closer relationships because is a smaller family for instance? Second, related to measurement, why steps? Is it significant in the context of energy expenditure?

Discussion 

Pg 9 Ln “To the best of our knowledge, this is the first study” I would rather the authors use the opening statement to summarize the study goals again. Then, present the main findings and expand upon in paragraphs thereafter.

Pg 10 272-279 Do you have an idea for this significant difference in weekday PA and weekend PA? What does the literature say? I would elaborate on this within this paragraph.

Pg 10 287-293 I agree with the authors that SES is complex, but there wasn’t a comprehensive effort by the authors to account for this in the design, nor in the methodology. Perhaps authors can speak on ways to improve in this area going forward.

Pg 10 Ln 307 “Better educated parents” replace with parents with greater educational attainment.

Author Response

We are very thankful for your comments about our article and the opportunity to revise our manuscript. We included all your helpful advice and made necessary changes in the article. Herein we explain how we revised the paper based on your comments and recommendations. Changes made in the manuscript are marked using track changes.

Abstract:

Comments for the Authors: The opening statement should be a bit stronger. Why is PA so important at an early age? Even more so “necessary”? Citing some specific benefits might be suitable here.

Author's response: The authors thank the reviewer for this comment. Now, a stronger opening statement is included in the abstract, with some examples of benefits.

Comments for the Authors: Pg 1 Ln 17-18: Here you state that you will assess the sample’s PA levels in relation to guideline recommendations. Which guidelines? Also, this should be a step in the context of the first aim (makes most sense).

Author's response: The authors thank the reviewer for this comment. We rewrote the aims of the study.

Comments for the Authors: Since the sample is mostly 5–6-year-olds, I would be more specific when referring to the group. Are these kindergarteners? First graders? In some countries, 5–6-year-olds are no longer considered preschoolers.

Author's response: We thank the reviewer for this comment. In Poland, children aged 3 to 6 attend kindergarten. From the age of 7 they go to primary school, therefore the whole study sample consisted of preschoolers.

Comments for the Authors: Pg 1 Ln 24-25: It is quite interesting that some of your early findings focus on a comparison between 5- and 6-year-olds. Were the authors originally intending to assess such comparisons? At first sight, I would think findings would be summarized for the total sample.

Author's response: Thank you for this comment. We intended so. Our previous analyses and studies confirmed the differences between 5 and 6 year olds. For example, 6-year-olds were more physical active.

Comments for the Authors: Pg 1 Ln 26: “genders.” Did you mean by biological sex of the child?

Author's response: Thank you for this remark. We changed the nomenclature related to sex in the text.

Comments for the Authors: Pg 1 Ln 32-38: The discussion provided elaborates on findings not presented. Where are the findings for study aim 2? Authors only presented findings related to guideline recommendations in the abstract. I encourage the authors to re-visit the abstract and be more specific about the indicators and models used to test each study aim.

Author's response: Thank you for this important comment. We rewrote the whole abstract section.

Introduction:

Comments for the Authors: “Physical activity (PA) during the preschool years is crucial for child development and 42 health as well as well-being [1,2].” Can the authors provide specific examples of developmental and health outcomes that benefit from PA engagement?

Author's response: We thank the reviewer for this comment. We rewrote the first paragraph of the introduction section accordingly.

Comments for the Authors: Pg 2 Ln 85-87 “Thus, focusing on the analysis of parent-child relationships in preschool children is the key to understanding the factors that are important in shaping children's active lifestyles [3,36].” Was there any assessment of parents’ PA engagement? Even through the questionnaire? When this statement was presented in the introduction, I thought that was where the authors were headed.

Author's response: Author's response: We thank the reviewer for this comment. The authors did not ask parents for their involvement in PA. The authors have corrected this sentence to make it clearer and not misleading.

Comments for the Authors: Pg 3 Ln 99-105 Same comment as in the abstract: Revise the aims, PA assessment with guidelines should fall under aim 1. Be specific about the vars and models in aim 2.

Author's response: We thank the reviewer for this comment. We corrected abstract and introduction according to the reviewer's indications.

Methods:

Comments for the Authors: Pg 3 Ln 111-113 “All the measurements were taken between 8:00 a.m. and 10:00 a.m. With the help of tutors/teachers, all participants received comprehensive information about the study.” How was the consent process? Did parents also receive such information? Did the parents provide informed consent? I am also curious about the 2-hour window of the measurement. Is it a validated approach?

Author's response: The authors thank the reviewer for this comment. Because the other reviewer also point this issue out we added relevant information in the first paragraph of subsection “Participants”. The consents, detailed information about the study and the questionnaires were delivered to the parents/caregivers through kindergarten staff. After signing parents left the documents in kindergarten. Because of the strict schedule, the head of kindergarten allowed only for 2 h for the measurements, therefore it can’t be stated that it was a validated approach.

Comments for the Authors: Pg 3 Ln 117 “healthy preschoolers.” Please define.

Author's response: The authors thank the reviewer for this comment. Since we had to rewrite the whole paragraph, we included this information as an exclusion criteria at the end of “Participants” paragraph, as below: “ Exclusions criteria were: (1) conditions that could affect the PA”.

Comments for the Authors: Pg 4 Ln 150-151: How many met criteria for a valid 7-day period of accumulated data? The total sample?

Author's response: The authors thank the reviewer for this comment. As we stated in rewrited subsection “Participants”, the 18 subjects did not meet the criteria for a valid 7-day period of accumulated data (“Of these, 43 were excluded from the study for the following reasons: strong anxiety of examination (n = 5), failure to return or complete the survey (n = 20), or lack of valid accelerometer data (n = 18). The final study group consisted of 522 children.”)

Results

Comments for the Authors: Table 1: The cut-offs for SES and education do not provide insight into how these variables may play a role in your outcome of interest. Perhaps it might be worth considering other theoretically relevant categorizations of SES if relevant to your research question.

Author's response: The authors thank the reviewer for this comment. The questionnaire used in the study unfortunately covered only general information related to household income (the possible answers related to household income was: 1.High; 2.Middle; 3.Low). Because only 5 families declared low and 35 families middle level of household income we decided to merge this two categories into Middle or Low household income (n=40). A similar approach was applied to education categories (4 mothers and 6 fathers declare primary education, while 116 mothers and 221 fathers declare secondary/middle education).

Comments for the Authors: Table 2: You have some really interesting findings related to weekday vs weekend PA. That’s the first time in the paper I read this. Should be presented in the abstract (it is not there currently).

Author's response: We agree with the reviewer. We rewrote the whole abstract section and included the findings from table 2.

Comments for the Authors: The findings about the exact number of people in the household (2) associated with increased odds of meeting guideline recommendations is interesting. First, why 2 people? Is this perhaps a matter of individuals occupying space in the household, while also having closer relationships because is a smaller family for instance? Second, related to measurement, why steps? Is it significant in the context of energy expenditure?

Author's response: We thank the reviewer for this comment. In the questionnaire, we assumed that two people are otherwise one parent and one child. We rewrote the relevant part of the discussion section to wider elaborate this association. Since many authors include the number of steps in the PA analysis – we decided to include this variable to extend our analysis. Despite there are also studies in literature which connect the number of steps with energy expenditure we were not aiming to analyse this associations in the context of present study.

Discussion 

Comments for the Authors: Pg 9 Ln “To the best of our knowledge, this is the first study” I would rather the authors use the opening statement to summarize the study goals again. Then, present the main findings and expand upon in paragraphs thereafter.

Author's response: We agree with the reviewer. We rewrote the opening paragraph of the discussion section accordingly.

Comments for the Authors: Pg 10 272-279 Do you have an idea for this significant difference in weekday PA and weekend PA? What does the literature say? I would elaborate on this within this paragraph.

Author's response: We thank the reviewer for this comment. We extended the relevant part of discussion to the possible explanations of significant difference in weekday PA and weekend PA based on the findings from literature.

Comments for the Authors: Pg 10 287-293 I agree with the authors that SES is complex, but there wasn’t a comprehensive effort by the authors to account for this in the design, nor in the methodology. Perhaps authors can speak on ways to improve in this area going forward.

Author's response: We thank the reviewer for this comment and we agree with the reviewer. Due to the incorrect translation in previous version of manuscript, the socio-economic status (SES) term was incorrectly used. Therefore in revised manuscript we had changed the term SES in to „selected socio – economic indicators”. In the present study the selected socioeconomic indicators were: place of residence, education of the parents, number of people in the household and household income. Therefore at the end of the discussion section we indicated what the future studies should incorporate in the study design.

Comments for the Authors: Pg 10 Ln 307 “Better educated parents” replace with parents with greater educational attainment

Author's response: We thank the reviewer for this remark. We changed this phrase accordingly.

Reviewer 3 Report

The aim of the study was 1) to assess the level of intensity and duration of PA in preschool girls and boys on weekdays and weekends;  2) to investigate the relationship between socio – demographic factors and PA. The aims of the study are important for PA promotion and public health. However, this study has methodological flaws that seriously compromise this research.

Materials and Methods. First, study organization must be presented comprehensively explaining how it was organized (how random selection of kindergartens was implemented, who, how organized testing of children). Secondly, the involvement of parents was unclear – was one of both parents invited to participate in study? How many of parents were women and men, what was their age? How data from parents was merged with data of children, what was done in case of two children in one family? Authors state (p. 4, line 145), that race of children was assessed, however no data about distribution of race was presented.

Authors claim that socio-economic status (SES) of parents was assessed, nevertheless it is unclear how it was done. No questions except of education of parents were included in questionnaire (or they are not presented). It is also unclear how and when parents filled questionnaires? Who and how organized this part of the study?

Age and gender of participating children should be included in Methods section Participants subsection.

The relative number of men and women is unclear, thus it is confusing when comparisons between mothers and fathers in education were described (p. 5, line 196).

Effect sizes in differences between study variables should be provided.

What are the measures for Urban/Rural or other variables (except of body weight/height) in Table 1?  Is it number (%)?

The statistics of Table 2 is confusing. How the p value was assessed for various intensity of PA in total sample? How was p value assessed for girls and boys? Were the differences for girls and boys or within what variables?

Table 4 is also confusing. Table 4 shows that 45.3% of girls and 55.4% for boys that meet MVPA recommendations in weekdays, meet them on weekends either. The same is for meeting criteria for steps (44.4%). However, the interpretation of results of the Table 4 says something very different (see p. 7, lines 226-235). Authors state that there is difference between boys and girls, however, statistics show no significant differences between genders.

Since it is confusion with methods, statistics and interpretation of study results, discussion and conclusions are doubtful. Sample of the study was relatively large, but it could not be stated that study was implemented in “relatively large population” (see conclusions, p. 11, line 333).

Rationale for the study should be more focused and evidence based. I suggest to have more focus on findings between parents’ SES, educational status and children PA. It is not enough just to state that findings are inconsistent (p. 2, line 76), it is important to present and analyze these inconsistencies.  

I believe that conclusions of the study are important for public health, however, the methods, statistics, results and interpretation of results compromise this research. 

Author Response

We are very thankful for your comments about our article and the opportunity to revise our manuscript. We included all your helpful advice and made necessary changes in the article. Herein we explain how we revised the paper based on your comments and recommendations. Changes made in the manuscript are marked using track changes.

Comments for the Authors: Materials and Methods. First, study organization must be presented comprehensively explaining how it was organized (how random selection of kindergartens was implemented, who, how organized testing of children). Secondly, the involvement of parents was unclear – was one of both parents invited to participate in study? How many of parents were women and men, what was their age? How data from parents was merged with data of children, what was done in case of two children in one family? Authors state (p. 4, line 145), that race of children was assessed, however no data about distribution of race was presented.

Author's response: Thank you for your suggestions. Below we explain the selection process for the study group. Based on data published by Polish Central Statistical Office there were around 6,800 preschoolers in years 2017-2018 in Rzeszów, Poland. Assuming a confidence level of 95%, and 5% margin of error the required sample size should cover at least 364 participants. The invitation to participate in the study was sent to all kindergartens in Rzeszów. The consents of 41 kindergartens’ principals were obtained for participation in this study. Based on data from the Rzeszów City Hall, the average number of preschoolers attending these kindergartens was around 70. Considering that approximately 30% of parents agree for child's participation in the study, and possible complications during examinations (missing data in surveys, failure to meet valid days when measuring PA with accelerometers, absence of children on the day of the test, etc.), we decided to randomly select 22 kindergartens to participate in the research (kindergartens were selected using STATISTICA software – sampling without replacement). The consent of 565 parents was obtained for child participation in the measurements for the purpose of this study. Of those respondents, 43 were excluded from the study due to failure to meet the eligibility criteria. The final study group consisted of 522 children.

The assumption was that the questionnaire was completed by one of the parents, but we do not have feedback on which parent (and whether both) completed the questionnaire. We did not collect information on the age of the parents. The questionnaire for the parents of the studied children was provided with an individual code that was also given to the children during the study. On this basis, the results of the questionnaire were identified with the results of the assessment of children's physical activity. In case of two children from one family, both children received their own code and the parent received two separate questionnaires to be completed.

No data about distribution of race was presented, due to the fact that all study group consisted of Caucasian race (completed in lines 141-169).

Comments for the Authors: Authors claim that socio-economic status (SES) of parents was assessed, nevertheless it is unclear how it was done. No questions except of education of parents were included in questionnaire (or they are not presented). It is also unclear how and when parents filled questionnaires? Who and how organized this part of the study?

Author's response: Thank you for this comment. Due to the incorrect translation in previous version of manuscript, the socio-economic status (SES) term was incorrectly used. Therefore in revised manuscript we had changed the term SES in to „selected socio – economic indicators”. In the present study the selected socioeconomic indicators were: place of residence, education of the parents, number of people in the household and household income. Paper and pencil questionnaire was delivered to subjects parents after receiving the consent for participation in the study. The questionnaires were delivered through kindergarten staff.

Comments for the Authors: Age and gender of participating children should be included in Methods section Participants subsection.

Author's response: Thank you this comment. Age and gender of participating children are provided in Participants subsection.

Comments for the Authors: The relative number of men and women is unclear, thus it is confusing when comparisons between mothers and fathers in education were described (p. 5, line 196).

Author's response: We thank the reviewer for this comment. The assumption was that the questionnaire was completed by one of parents, but we do not have feedback on which parent (and whether both) completed the questionnaire. However, in the surveys we have received, we have information about the mother and father of each child (there are 522 results for the mother and 522 results for the father) (table 1).

Comments for the Authors: Effect sizes in differences between study variables should be provided.

Author's response: We thank the reviewer for this comment. Differences between study variables has been provided in table 2.

Comments for the Authors: What are the measures for Urban/Rural or other variables (except of body weight/height) in Table 1?  Is it number (%)?

Author's response: Yes – the measures for Urban/Rural or other variables are the numbers (%). Data marked with an asterisk (*) mean number (%), what has been explained in footnote under the table.

Comments for the Authors: The statistics of Table 2 is confusing. How the p value was assessed for various intensity of PA in total sample? How was p value assessed for girls and boys? Were the differences for girls and boys or within what variables?

Author's response: Thank you for this comment. In Table 2 we assess the differences in individual PA levels (light, moderate, etc.) between the weekend and weekdays in three separate groups: total, girls, and boys. The p value was assessed in the same method in total sample and in girls and boys groups. In table 2 we do not analyse (with p value) differences in PA levels between girls and boys but only between weekend and weekdays in seperate groups (total sample, boys, girls).

Comments for the Authors: Table 4 is also confusing. Table 4 shows that 45.3% of girls and 55.4% for boys that meet MVPA recommendations in weekdays, meet them on weekends either. The same is for meeting criteria for steps (44.4%). However, the interpretation of results of the Table 4 says something very different (see p. 7, lines 226-235). Authors state that there is difference between boys and girls, however, statistics show no significant differences between genders.

Author's response: We thank the reviewer for this important comment. The description of the results in Table 4 has been thoroughly revised.

Comments for the Authors: Since it is confusion with methods, statistics and interpretation of study results, discussion and conclusions are doubtful. Sample of the study was relatively large, but it could not be stated that study was implemented in “relatively large population” (see conclusions, p. 11, line 333).

Author's response:  The authors corrected the information in the section: methods, statistics and interpretation of  results. The sentence “relatively large population” has been removed.

Comments for the Authors: Rationale for the study should be more focused and evidence based. I suggest to have more focus on findings between parents’ SES, educational status and children PA. It is not enough just to state that findings are inconsistent (p. 2, line 76), it is important to present and analyse these inconsistencies.  

Author's response: We thank the reviewer for this important comment. In the Introduction the authors supplemented the information by focusing more on the findings of selected socioeconomic indicators and PA of children.

Again, we appreciate all your insightful comments and the opportunity to revise our paper. Thank you for taking the time and energy to help us improve the paper. We hope our revision meets your approval.

Yours faithfully,

Authors

Round 2

Reviewer 1 Report

The authors have addressed all my comments and concerns well. I appreciate all their effort

Author Response

We are very thankful for your comments about our article and given the opportunity to revise our manuscript

Reviewer 3 Report

Authors addressed some my comments, however, there were left some important aspects that need additional work.

First, authors developed assumption that is not associated with the main aims of the study. Was the aim to compare PA in kindergarten and at home? Aims were related to comparison of PA in weekdays and weekends, thus, the assumption should ne aligned with it. The same for the first aim – meeting WHO recommendations. What was hypothesis for this? Please align the aims of the study, hypotheses, and introduction. It should be clear from the introduction why it is important to answer research questions and what is the possible assumptions in line with the literature.

Unfortunately, the procedure of testing parents and children is not described properly in Methods. Please include the subsection Procedure and explain 1) the procedure of sample selection and 2) procedures of questioning of parents and 3) procedures of testing children providing specific information (who and how implemented this). Participants’ subsection in Methods should be revised.

There are still unclear what questions were used to get answers for selected sociodemographic indicators? Please provide these questions and reference proving the validity of these questions.

Please provide information in Statistical analyses what kind of partial η2 eta’s were provided for the effect size, what are the ranges for low, average and high effect sizes.

I should recommend revising the name of the article as: The associations between PA of preschool children and selected socioeconomic indicators of family.

It should be clear from the aim of the study that research is implemented in a sample of Polish preschool children and their parents.

Much work must be done in Introduction. Authors included some important aspects however the novelty of the research should be described deeper. The lack of studies in Polish preschool children is important for public health, bet it should be clear what new this study adds for a global public health science.

I respectably disagree that we can measure the BMI or education of one of the parents and make conclusions about the associations between parental education and PA of kids. As we have information from only one of parents, how can we state that there are no associations?  I would suggest using only these sociodemographic variables in the present study: 1) income of the family; 2) no of people on the household; 3) place of residence. If authors would decide to keep all variables, previously mentioned aspect should be discussed as the limitation of the study.

Age range of the sample for girls and boys should be included in Abstract. Since study is cross -sectional, please avoid causal language (see conclusions in the abstract).

Author Response

We are very thankful for your comments about our article and given the opportunity to revise our manuscript. We included all your helpful advice and made necessary changes in the article. Herein we explain how we revised the paper based on your comments and recommendations. Changes made to the manuscript are marked in blue.

Authors addressed some my comments, however, there were left some important aspects that need additional work.

First, authors developed assumption that is not associated with the main aims of the study. Was the aim to compare PA in kindergarten and at home? Aims were related to comparison of PA in weekdays and weekends, thus, the assumption should ne aligned with it.

The same for the first aim – meeting WHO recommendations. What was hypothesis for this? Please align the aims of the study, hypotheses, and introduction. It should be clear from the introduction why it is important to answer research questions and what is the possible assumptions in line with the literature.

Author's response: The authors thank the reviewer for this comment. We agree with the reviewer  that in previous version of the manuscript the rationale for the stated assumption (hypothesis) was not enough consistent along with the introduction and aims of the study. The aim of the study was not to compare the PA in kindergarten and at home but on weekdays and at the weekends, therefore we changed the wording throughout the paper to be more consistent with the study aims. Since there are no consensus in outcomes in literature according to PA level during weekdays and weekend - we changed the hypothesis to be more align with literature findings and study aims.

Unfortunately, the procedure of testing parents and children is not described properly in Methods. Please include the subsection Procedure and explain 1) the procedure of sample selection and 2) procedures of questioning of parents and 3) procedures of testing children providing specific information (who and how implemented this). Participants’ subsection in Methods should be revised.

Author's response: The authors thank the reviewer for this suggestion. We rewritten the Methods section according to the reviewer suggestions.

There are still unclear what questions were used to get answers for selected sociodemographic indicators? Please provide these questions and reference proving the validity of these questions.

Author's response: The authors thank the reviewer for this question. We included the questionnaire used (translated to English) as a supplementary material (suppl. 1). All the questions used in questionnaire was developed by authors therefore we do not have the data on the validity of questionnaire, what should be considered as an another limitation of the study. We added relevant information in the limitation subsection.

Please provide information in Statistical analyses what kind of partial η2 eta’s were provided for the effect size, what are the ranges for low, average and high effect sizes.

Author's response: The authors thank the reviewer for this remark. We added relevant information and ranges for interpretation of partial η2 eta’s values in the statistical analyses subsection.

I should recommend revising the name of the article as: The associations between PA of preschool children and selected socioeconomic indicators of family.

Author's response: The authors thank the reviewer for this remark. Taking into account that the main aim of study was to compare the PA during weekdays and at weekend and the assessment of associations between PA and selected socioeconomic indicators of family was secondary aim of the study we respectably we would like to remain with the current title of our manuscript.

It should be clear from the aim of the study that research is implemented in a sample of Polish preschool children and their parents.

Author's response: The authors thank the reviewer for this suggestion. We added relevant information in the aims both in abstract and at the end of introduction section.

Much work must be done in Introduction. Authors included some important aspects however the novelty of the research should be described deeper. The lack of studies in Polish preschool children is important for public health, bet it should be clear what new this study adds for a global public health science.

Author's response: The authors thank the reviewer for this comment. We included in the introduction the information regarding novelty of our research.

I respectably disagree that we can measure the BMI or education of one of the parents and make conclusions about the associations between parental education and PA of kids. As we have information from only one of parents, how can we state that there are no associations?  I would suggest using only these sociodemographic variables in the present study: 1) income of the family; 2) no of people on the household; 3) place of residence. If authors would decide to keep all variables, previously mentioned aspect should be discussed as the limitation of the study.

Author's response: The authors thank the reviewer for this comment. Answering previous comment of the reviewer we clarified in the revised manuscript the procedure of testing parents. The translated questionnaire was also included as supplementary material. For each children included to study separate questionnaires were delivered. The questionnaire contain the questions regarding mother and father body height, body mass and education level ect. The parents were instructed to discuss and complete the questionnaire together – compliting appropriate information for each of parent. Therefore we decided to keep all analysed variables. However, the lack of validation of used questions should be considered as limitation of study (as we stated above).

 Age range of the sample for girls and boys should be included in Abstract. Since study is cross -sectional, please avoid causal language (see conclusions in the abstract).

Author's response: The authors thank the reviewer for this comment. We clarified the age range of the sample for girls and boys in abstract. We also replaced the causal language throughout the manuscript with wording appropriate for the cross -sectional nature of study.